# Quality of Life in Women Subjected to Surgical Treatment of Breast Cancer Depending on the Procedure Performed within the Breast and Axillary Fossa—A Single-Center, One Year Prospective Analysis

**DOI:** 10.3390/jcm10071339

**Published:** 2021-03-24

**Authors:** Magdalena Tarkowska, Iwona Głowacka-Mrotek, Tomasz Nowikiewicz, Aleksander Goch, Wojciech Zegarski

**Affiliations:** 1Department of Physiotherapy, Nicolaus Copernicus University in Toruń, Collegium Medicum in Bydgoszcz, 85-094 Bydgoszcz, Poland; magdalena.sowa@cm.umk.pl (M.T.); a.goch@cm.umk.pl (A.G.); 2Department of Rehabilitation, Nicolaus Copernicus University in Toruń, Collegium Medicum in Bydgoszcz, 85-094 Bydgoszcz, Poland; 3Department of Surgical Oncology, Nicolaus Copernicus University in Toruń, Collegium Medicum in Bydgoszcz, 85-094 Bydgoszcz, Poland; tomasz.nowikiewicz@cm.umk.pl (T.N.); zegarskiw@cm.umk.pl (W.Z.); 4Department of Clinical Breast Cancer and Reconstructive Surgery, Franciszek Łukaszczyk Oncology Center, Romanowskiej Street, 85-796 Bydgoszcz, Poland

**Keywords:** breastcancer, quality of life, limfadenectomy, sentinel lymph node biopsy

## Abstract

The aim of this study was to evaluate the quality of life of patients undergoing surgical treatment of breast cancer depending on the type of procedure involving the breast (mastectomy vs. breast conserving treatment) and axillary fossa (sentinel lymph node biopsy vs. axillary lymph node dissection). The prospective study was carried out in a group of 338 females undergoing breast cancer treatment. Study variables were assessed by means of a diagnostic survey using standardized QLQ C30 and BR23 questionnaires as well as the Acceptance of Illness Scale and Mini-MAC scales. The quality of life was assessed at threetime points: on the day before the surgical procedure (I assessment) as well as three and 12 months after surgery (II and III assessment). Statistically significant differences between study groups were observed in the overall quality of life subscale (I, II, III—*p* < 0.0001), physical functioning (I—*p* < 0.0001; II—*p* = 0.0413; III—*p* < 0.0001), role functioning (I—*p* = 0.0002; III—*p* < 0.0001), emotional functioning (III—*p* = 0.0082), cognitive functioning (I—*p* = 0.0112; III—*p* < 0.0001), social functioning (III—*p* < 0.0001), body image (I, II, III—*p* < 0.0001), and sexual functioning (I—*p* = 0.0233; III—*p* = 0.0011). In most symptomatic scales, significant (*p* < 0.05) differences were also noted. Mastectomy and limfadenectomy patients were significantly (*p* < 0.0001) more prone to present with destructive coping strategies one year after surgery. Breast conserving therapy is associated with better quality of life outcomes as compared to mastectomy. Sentinel lymph node biopsy is associated with a lower intensity of adverse changes in multiple dimensions of patients’ functioning.

## 1. Introduction

Breast cancer is the most prevalent malignancy in women in Poland and highly developed countries [1]. The treatment of breast cancer is referred to as combination treatment. Two surgical modalities are available, including mastectomy and breast conserving treatment (BCT); with regard to the local lymphatic system, surgeries may involve axillary lymph node dissection (ALND) or sentinel lymph node biopsy (SLNB). The mainstay of the therapeutic regimen consists in surgical procedure supplemented by various adjuvant treatments [2].

Surgical interventions within the mammary gland result in physical mutilation as well as in psychological and social trauma. Attention to the patient’s quality of life is considered an important element of holistic treatment approach. The diagnosis of malignancy may trigger a number of negative emotions. The quality of life is determined by such factors as acceptance of illness and adjustment to the new health status. The acceptance of cancer is particularly challenging as it requires acknowledging the presence of malignancy along with the negative impact of treatment on many aspects of daily functioning [3,4].

The aim of this study was to carry out a prospective, single-center analysis of the quality of life depending on the type of breast and axillary surgery in one-year follow-up of patients having undergone surgical treatment for breast cancer.

## 2. Material and Methods

The study followed a prospective, single-center, three-stage pretest-posttest observation design. The study had received the approval of the Bioethics Committee of the Nicolaus Copernicus University in Toruń, decision no. 641/2015. The study started in November 2015 and was completed in May 2018. The per-analysis group consisted of 338 patients, including 153 patients subjected to mastectomy and 185 patients subjected to BCT.

Inclusion criteria:Written consent to participate in the study.Age of above 18.Hospitalization for surgical treatment of breast cancer at the Clinical Department of Breast Cancer and Reconstructive Surgery of the Oncology Centre in Bydgoszcz during the first stage of the study (November 2015–May 2017).Overall good performance status (ECOG 0-1).Exclusion criteria:Other severe comorbidities (above ASA II).Treatment modality being changed from BCT to mastectomy during the study.Distant metastases at the time of qualification for the study.Breast reconstruction surgery during the study.Lack of consent for breast-conserving treatment (according to patient’s preferences).Other malignancy diagnosed in the subject within the last 5 years.

In the course of our scentific research, it has been decided that patients who undergo breast reconstruction surgery after mastectomy should be excluded from this analysis. During this study, 33 patients underwent breast reconstruction surgery. Only 3 of these patients had simultaneous breast reconstruction, while the other 30 underwent this procedure at various times after their initial tumor removal surgery.

All patients included in this study were subjected to radical surgical treatment. According to current guidelines, it was necessary to remove a tumor within healthy tissue margins (no ink on tumor). Surgical procedures performed on primary tumors included local removal of neoplastic lesions and preservation of healthy tissue margins—breast conserving treatment (BCT). The width of obtained tumor margin was then confirmed on histopathological examination. However, if primary tumors exceeded 3–4 cm in their longest diameter (which prevented surgeons from achieving a satisfactory aesthetic effect of BCT) or if multifocal neoplastic lesions were present, patients were qualified for mastectomy. Scheme describing patient exclusion from the study are shown in Figure 1. 

The quality of life was assessed 3 times. Assessment I was held on the day before the surgery after successive patients reporting at the Department had been qualified according to the inclusion and exclusion criteria following examination of their medical records. Assessments II and III were carried out by means of computer-assisted telephone interviewing (CATI) 3 and 12 months after surgery, respectively.

In order to examine demographic variables, a proprietary questionnaire was designed including questions regarding patients’ age, educational background, residence, employment status, marital status and socioeconomic status.

Clinical parameters were determined by means of medical documentation analysis. Parameters determined for the purposes of statistical analysis included the operated side, neoadjuvant treatment, clinical staging, adjuvant treatment, estrogen and progesterone receptor status, and HER2 status.

Standardized research tools used to assess quality of life, acceptance of illness and mental adjustment to cancer included:The EORTC QLQ-C30 questionnaire, version 3.0: an international, standardized test tool developed by the European Organisation for Research and Treatment of Cancer (EORTC) to assess patients’ physical, social, and emotional functioning, role-based performance, memory and concentration, fatigue, pain, nausea and vomiting, loss of appetite, diarrhea, constipation, financial difficulties, dyspnea, insomnia, and overall quality of life. The QLQ-C30 a self-administered tool used in all patients diagnosed with cancer regardless of cancer stage, type, and location.The EORTC QLQ-BR23 questionnaire for more accurate assessment of the quality of life of women with breast cancer; the questionnaire consists of two functional status scales (body image and sexual functioning), three symptomatic scales (systemic therapy side effects, arm symptoms, and discomfort originating from the operated breast), and three questions regarding patient’s concerns about hair loss, future perspective, and sexual enjoyment. Results obtained for each subscale of QLQ-C30 and QLQ-BR23 questionnaires are in the range of 0–100. For functional scales, higher results mean better performance within the particular aspect whereas for symptomatic scales, the higher the result, the greater the severity of the condition and the poorer the quality of life.The Acceptance of Illness Scale (AIS) as developed at the Center for Community Research and Action, Department of Psychology, New York University. The scale consists of eight statements relating to the consequences of ill health. Individual claims pertain to illness-related restrictions, reduced self-esteem, and lack of self-sufficiency. A higher degree of acceptance contributes to faster adjustment to the new health condition and to a better quality of life. The scale is intended for adult patients regardless of their disease. The score range is 8–40 points. The higher the score, the better the acceptance of current illness and the less negative emotions associated with it.The Mini-Mental Adjustment to Cancer (Mini-MAC) scale consists of 29 statements that measure four methods of coping with cancer, including anxious preoccupation, fighting spirit, helplessness-hopelessness, and positive re-evaluation. The first two strategies account for a constructive coping style whereas the last two comprise a destructive style of coping with the disease. The scale can be used to assess patients’ response to the diagnosis of cancer as well as their adaptation to subsequent treatment and rehabilitation stages. The coping strategy may be a measure of the quality of life. Higher scores are indicative of a predominant contribution of a particular coping style.

### Statistical Analysis

Statistical analyses were carried out using the PQStat statistical package, version 1.6.4.110 (PQStat Software, Poznan, Poland). Qualitative patient characteristics and types of axillary surgeries in BCT and MAS groups were analyzed using the chi-square test for independence. The results of the QLQ-C30, QLQ-BR23, Mini-MAC, and AIS questionnaires in BCT + ALND, BCT + SNB, MAS + ALND, and MAS + SNB groups were analyzed using the Kruskal–Wallis test and the Dunn–Bonferroni post-hoc test. The probability value of *p* < 0.05 was considered significant while the probability value of *p* < 0.01 was considered highly significant.

## 3. Results

The study groups were characterized in terms of clinical and demographic data. No significant differences (*p* > 0.05) were observed between MAS and BCT groups with regard to the frequency of adjuvant chemotherapy, operated breast side and family history of breast cancer. However, the BCT and MAS groups differed significantly in terms of other clinical variables. Patients qualified for mastectomy were more frequently treated with neoadjuvant modalities (*p* < 0.0001) and immunotherapy (*p* = 0.0104). Higher cancer stages (*p* < 0.0001) and positive HER2 receptor status (*p* = 0.0104) were also significantly more frequent in this group. Patients undergoing breast-sparing surgery were more likely to be treated with adjuvant endocrine therapy (*p* = 0.0225) and radiotherapy (*p* < 0.0001). A higher percentage of postmenopausal women (*p* = 0.0157) as well as a significantly higher percentages of patients with positive estrogen receptor status (*p* = 0.0225) and progesterone receptor status (*p* = 0.0014) in cancer cells were also observed more frequently in the BCT group.

No significant (*p* > 0.05) differences were identified between the BCT and MAS groups in terms of demographic variables, i.e., age, educational background, area of residence, marital status, or socioeconomic status. Details are presented in Table 1.

Study patients were also characterized in terms of the type of surgical procedure within the regional lymphatic system. A highly significant relationship (*p* < 0.0001) was identified between the type of surgery and the axillary procedure. SLNB was more common in patients qualified for breast saving surgery whereas ALND was more common in patients qualified for mastectomy. Details are presented in Table 2.

The next step consisted in the comparison of the quality of life results obtained using the QLQ-C30 questionnaire before the procedure as well as threeand 12 months after the procedure depending on the type of breast and axillary surgery. Detailed results related to the physical, social, emotional, and cognitive functioning, role-based performance and the overall quality of life and disease symptoms are presented in Scheme 1 Statistically significant (*p* < 0.05) differences in results were reported for the overall health and quality of life as well as physical functioning scales at all time points. Regardless of the time point, the lowest results indicative of the worst quality of life were recorded in the MAS + ALND group. In all groups, the lowest results were observed at the second time point. Results in patients undergoing mastectomy were lower than those in patients subjected to BCT. At each stage of the study, patients undergoing axillary lymph node dissection presented with lower scores suggestive of lower overall quality of life and physical functioning as compared to patients subjected to sentinel lymph node biopsy. No significant differences (*p* > 0.05) were observed in emotional functioning between subgroups at the first and the second time point. A highly significant difference (*p* = 0.0082) was observed at the third time point with the lowest results corresponding to the most difficult emotions being reported in the MAS + ALND group. The best emotional functioning at the third time point was observed in the BCT + SNB group. The results suggest poorer emotional functioning of patients qualified for mastectomy as compared to BCT as well as patients qualified for ALND as compared to SLNB. A systematic increase in scores was observed at subsequent study time points indicating that emotional functioning improved with time from the surgery. Statistically significant differences were also observed in the role-based performance subscale at time points I (*p* = 0.0002) and III (*p* < 0.000001), the cognitive functioning subscale at time points I (*p* = 0.0112) and III (*p* < 0.00001) as well as the social functioning subscale at time point III (*p* < 0.00001). Similarly, BCT vs. mastectomy and axillary node-saving surgery (regardless of the type of breast surgery) facilitated better functioning and a higher quality of life.

With regard to the symptomatic scales, statistically significant differences in results were observed at time points I and III for the subscales of fatigue (I: *p* < 0.0001, II: *p* < 0.0001), nausea and vomiting (I: *p* = 0.0077, II: *p* = 0.0201), and pain (I: *p* = 0.0091, II: *p* < 0.0001), at time point III for the subscales of dyspnea (*p* = 0.0017), appetite loss (*p* = 0.0005) and financial difficulties (*p* < 0.0001) as well as at time point I for the subscale of diarrhea (*p* = 0.0433). The highest values were recorded in the MAS + ALND group, which meant the strongest intensity of symptoms experienced by patients in this group, standing out from the other study participants. The results indicate that patients qualified for mastectomy were more likely to report adverse reactions compared to patients subjected to BCT. In addition, ALND increased the intensity of treatment-emergent symptoms. No statistically significant differences were observed for the scales of insomnia and constipation (*p* > 0.05).

The results obtained using the QLQ-BR23 questionnaire are presented in Scheme 2. The results in the body image subscale varied significantly (*p* < 0.0001) between the groups at all three assessment time points. At all time points, the quality of life as assessed using this scale was found to be better in the BCT group. At time points I and II, the highest scores reflecting the best self-image were recorded in the BCT + SLNB group. The results in the sexual functioning subscale varied significantly (*p* = 0.0233) between the groups at the first assessment time point. The lowest results corresponding to the lowest quality of life as assessed using this subscale were recorded at this stage in the BCT + SLNB. Highly significant (*p* = 0.0011) differences were also observed at the third assessment time point, with the highest values being—quite interestingly—reported in the MAS + ALND subgroup.

The greatest inter-group differences were observed in symptomatic scales. Statistically significant differences in results were observed in the subscales assessing the side effects of therapy in (time points I *p* < 0.0001 and III *p* = 0.0068), breast symptoms (time point III *p* = 0.008), and arm symptoms (time points II *p* < 0.0001 and III *p* < 0.0001). Regardless of the type of breast surgery, the need for axillary lymph node dissection significantly reduced the self-perceived quality of life in these subscales. The greatest severity of symptoms measured using these subscales was observed in the MAS + ALND group. The results in the body image subscale showed highly significant differences between groups at the study assessment time point III (*p* = 0.0002). The highest results were reported for the MAS + ALND group reflecting a greater concern regarding future health in this group as compared to the BCT + SLNB group, where the results were the lowest. Irrespective of the type of surgery within the breast and the lymphatic system, the highest results were reported prior to the surgery, whereas the lowest results were reported after one year. This indicates that with time, patients tended to perceive their perspectives as more favorable. No statistically significant differences were observed between groups in the subscales of sexual enjoyment and hair loss-related concern (*p* > 0.05).

The results obtained using the AIS questionnaire are presented in Table 3. Significant statistical differences (*p* < 0.0001) were observed between groups at each assessment time point. In each case, better acceptance of illness was observed in the BCT group as compared to the mastectomy group. Regardless of the type of breast surgery, the need for axillary lymph node dissection reduced the acceptance of illness.

Table 4 presents the results obtained using the Mini-MAC scale. With regard to the constructive coping styles subscale, statistically significant differences between groups were observed at assessment points I and II. Regardless of the type of breast surgery, the sentinel lymph node biopsy procedure was associated with higher contribution of constructive coping strategies as compared to axillary lymph node dissection. With regard to the destructive coping styles subscale, a statistically significant difference between groups was observed 12 months after the procedure. The need for breast amputation vs. BCT and the need to perform ALND vs. SLNB contributed to higher contributions of destructive coping strategies.

## 4. Discussion

This study assessed the quality of life of patients treated for breast cancer depending on the type of surgery performed within the breast and axillary region over a 12-month follow-up period. Standardized test tools were used for this purpose, including QLQ-C30, QLQ-BR23, AIS, and MINI-MAC scales. The strengths of this study included its prospective design, large population, and single center setting which increased the reliability of results.

The results of our studies are indicative of the quality of life being significantly lower in many functional aspects in patients subjected to mastectomy as compared to patients undergoing BCT. In the former group, worse quality of life was observed in both symptomatic and functional subscales of QLQ-C30 and QLQ-BR23 questionnaires. This might be the consequence of the surgical intervention required in this group of patients being more invasive and thus physically mutilating and emotionally challenging.

The reduced quality of life of patients treated for breast cancer was confirmed in many recent scientific reports [5,6]. Both BCT and mastectomy are associated with the possibility of multiple functional disorders developing as the result of the treatment. King et al. demonstrated that breast conserving treatment leads to tangible benefits in the self-perceived quality of life, particularly among young, married women. This was mainly reflected in the patients’ self-perception of body image [6]. Another study revealed that, 12 months after surgery, better quality of life scores were reported by BCT patients with regard to physical and social functioning, sexual enjoyment, and body image [7]. Our study has confirmed these observations. On the other hand, Aertc et al. demonstrated that neither the technique nor the extent of the surgery had any impact on the self-perceived quality of life in women treated for breast cancer [8].

Cancer is associated with long-term stress, emotional tension and irritability and consequently leads to deterioration of patients’ emotional condition. The importance of psychooncological treatment is increasingly highlighted as necessary for recovery. Our study revealed that the lowest emotional functioning scores were reported on the day before the surgery, and a growing trend was observed at subsequent assessment time points. Similar results were reported by other researchers [8].

Other factors responsible for the reduced quality of life in breast cancer to patients may include adverse reactions related to diagnostic and therapeutic procedures. In our study, treatment-related adverse reactions were significantly more common in the mastectomy group. Numerous research studies confirm worse results being observed with regard to adverse reactions in patients after mastectomy as compared to patients undergoing BCT [9].

In our study, a change in self-perceived body image has also been observed in the mastectomy group. Other authors also confirm that patients may feel inferior, flawed, and anxious at having to return to their daily lives [8,10].

In our studies, lower results in numerous quality of life scales were reported in patients qualified for complete resection of the lipid-lymphatic contents of axillary fossa as compared to patients undergoing sentinel lymph node biopsy regardless of the type of breast surgery (mastectomy or BCT). ALND was found to have a negative impact on self-perceived overall quality of life, physical functioning, role-based performance, as well as emotional, cognitive, and social functioning. With regard to symptomatic scales, significant impact on the deterioration of quality of life was observed for fatigue, nausea and vomiting, pain, diarrhea, dyspnea, loss of appetite and financial difficulties resulting from disease. Worse results were also observed in sub-groups in which the entire lipid-lymphatic contents of axillary fossa had to be removed in the ALND procedure. Most frequently, impairment of the quality of life in these scales was observed in the MAS + ALND subgroup. Similar relationships were noted for the BR23 questionnaire: the need to undergo ALND procedure significantly reduced the quality of live in both functional and symptomatic scales. Significant differences were identified particularly in self-perceived body image and adverse treatment reactions. In most cases, symptoms of the highest severity were reported by patients subjected to mastectomy with axillary lymph node dissection. We observed better functioning in symptomatic scales in the BCT + SLNB subgroup. However, regardless of the type of breast surgery, the necessity to carry out ALND contributed to increased severity of adverse reactions within the breast and the upper limb on the operated side as well as to increaser severity of the side effects of the treatment. Similar observations had also been made by other authors [11]. In their prospective, observational pretest/posttest study, Belmonte et al. assessed the quality of life of patients after sentinel lymph node biopsy or axillary lymph node dissection. The measurements were made in the early pre-operative period as well as one month, six months and one year after the surgery. Although the impairment in the quality of life was observed regardless of the type of axillary intervention, it was greater in ALND patients as compared to the SLNB group (*p* = 0.009) [12]. A meta-analysis carried out by Li et al. showed that patients having undergone ALND were more likely to report upper limb discomfort compared to patients subjected to SLNB [13]. Other researchers observed a significantly higher severity of upper limb discomfort in patients after ALND as compared to the SLNB group (68% vs. 36%). However, no significant differences were observed in this study with regard to self-perceived body image or sexual enjoyment [14]. Kootstra et al. demonstrated that two years after the procedure, the functioning of patients having undergone SLNB improved to the preoperative level while the functioning of patients having undergone ALND, albeit increased, could not reach the preoperative level [15].

Another objective of our research was to evaluate the degree of illness acceptance; to this end, the standardized AIS questionnaire was used. The acceptance of illness is an individual process of adapting to new physical conditions. Many authors have highlighted the importance of acceptance as an important factor in the recovery process [4]. Our study revealed that the acceptance of cancer was better in patients in whom both the breast and the regional lymph nodes were preserved in the surgery. Other researchers noted that the higher the acceptance of the new health condition, the better the quality of life in all functional scales and the lower the severity of adverse treatment reactions [16]. Cieślak et al. reported on a relationship between the severity of depressive symptoms and the level of illness acceptance in breast cancer patients [17].

The last objective of our study was to assess patients’ mental adjustment to cancer. The standardized mini-MAC scale was used for this purpose. The disease coping style is determined primarily by personality traits and medical history and may change at various stages of the treatment process [18,19]. In our study, we observed that breast and axillary lymph node-sparing surgeries were associated with constructive coping strategies being chosen more frequently than in cases of more mutilating surgeries. Many studies have demonstrated the link between self-perceived quality of life and the strategies of adapting to cancer disease [18,20]. Religioni et al. studied the mental adjustment to cancer in 902 patients treated for lung, prostate, breast, and colorectal cancer. Active fighting approach was observed in patients treated for breast or colorectal cancer. The authors observed increased contribution of anxious preoccupation and helplessness-hopelessness strategies in breast cancer patients who had received chemotherapy within the last 12 months [18]. In addition, it was found that the coping styles may change at the different stages of treatment [18]. Many studies have demonstrated the link between self-perceived quality of life and the strategies of adapting to cancer disease [18,20].

Moreover, a study carried out by Deepa et al. provides us with even more interesting evidence. In a research conducted on women in India five years after surgery, the authors found no evidence of differences in the quality of life between patients who underwent BCT and mastectomy [21]. Other studies revealed better quality of life in terms of body image, as well as cognitive and role functioning, in patients subjected to BCT [22]. Research conducted on Polish patients who underwent either mastectomy or BCT procedures demonstrated no statistically significant differences in the quality of life at three years after surgery [23]. Other studies conducted with the use of WHOQOL-BREF questionnaire have proved that patients who underwent mastectomy reported worse functioning, especially in social and environmental aspects [24].

Our study, despite its high clinical relevance, had certain limitations. Firstly, no randomization was provided for in the study design. Secondly, emotional functioning results at first assessment time point may be biased due to patients being anxious and distressed on the day before surgery. Another limitation of the authors’ own research is the fact that it was a single-center study. Furthermore, our analysis of the patient’s quality of life did not include the type of administered chemotherapy.

## 5. Conclusions


Patients qualified for BCT presented with better self-perceived quality of life in many functional dimensions before surgery as well as threeand 12 months afterwards as compared to patients having undergone mastectomy.Axillary lymph node dissection contributed to reduced quality of life in numerous functional dimensions regardless of the type of breast surgery. With regard to most functional and symptomatic scales, the best quality of life was observed in women subjected to BCT sentinel lymph node biopsy while the worst quality of life was observed in patients subjected to mastectomy with axillary lymph node dissection.BCT with SLNB offered better acceptance of illness levels and contributed to patients choosing more constructive strategies for coping with breast cancer.


## Data Availability

The datasets generated during and/or analysed during the current study are available from the corresponding author on request.

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
