# Peer review of "Quality of Life in Women Subjected to Surgical Treatment of Breast Cancer Depending on the Procedure Performed within the Breast and Axillary Fossa—A Single-Center, One Year Prospective Analysis"

_jcm, 2021, doi:10.3390/jcm10071339_

Round 1
Reviewer 1 Report
The paper is purely descriptive and does not contribute new information to the field . Data is from a single center which limits the conclusions.
Suggest listing types of chemo as this would contribute to QOL during the first year The EORTC QLQ BR23 may address this
The reason for mastectomy should be clarified For ex if this was patient preference, this may effect the outcome of the study
Why was reconstruction excluded during the study
What was the reason for withdrawal from the study?
The tables are cumbersome ; is there another way to display the results perhaps in graph form
Lastly no one from Psychiatry collaborated with the study
Author Response
Dear Reviewer #1,
Thank you for reviewing our article titled ‘Quality of life in women subjected to surgical treatment of breast cancer depending on the procedure performed within the breast and axillary fossa — a single-center, one year prospective analysis’ We deeply appreciate your opinion as well as constructive comments that contributed to more profound consideration of issues addressed in our publication. The comments in your review will guide us in our future work.
In response to those commentaries we clarified the Materials and Methods section, where we described precisely the process of patient qualification to mastectomy procedure.
Although we agree with you that this scientific paper is a single-center study, the healthcare facility where we conducted our research is a leading center in breast cancer surgery in Poland. It is true that there are numerous articles discussing the quality of life of women who undergo surgical treatment for breast cancer - it has been a “trending” topic in the field for the past several years. However, our work stands out among other studies due to the fact that we introduced a year-long prospective evaluation of our patients. The fact that our research is a single-center study has been stated in the limitations of the study section.
We also agree with you that the type of chemotherapy could influence the quality of life in the first year after surgery. Unfortunately, we have not taken this factor into account while designing the protocol of the study, as it has not been our primary interest in this research. This information has also been included in the limitations of the study section.
Breast reconstruction surgery is one of the exclusion criteria mentioned in our research protocol. During this study, 33 patients underwent breast reconstruction surgery. Only 3 of those patients had simultaneous breast reconstruction. We have concluded that performing reconstruction procedures at various times after surgery could influence the reliability of our study.IBR (immediate breast reconstruction) is a very rare procedure in Poland. Some reasons why Polish women resign from breast reconstruction are: fear of another surgical intervention, fear of surgical complications and hospitalization, age, fear of disease recurrence, as well as economic issues. Women are also inadequately informed about the course of breast reconstruction surgery and its consequences. Some patients do not decide for this procedure due to their advanced age and the fact that breast reconstruction may be perceived as an act of vanity.
Out material and methods section now features a more precise description of the exclusion criteria. We have also provided more details on the eligibility criteria for mastectomy procedure: all patients enrolled in this study underwent radical surgical treatment. According to current guidelines, it was necessary to remove a tumor within healthy tissue margins (no ink on tumor). Surgical procedures performed on primary tumors involved local removal of neoplastic lesions and preservation of healthy tissue margins - breast conserving treatment (BCT). The width of obtained tumor margin was then confirmed on histopathological examination. However, if primary tumors exceeded 3-4 cm in their longest diameter (which prevented surgeons from achieving a satisfactory aesthetic effect of BCT) or if multifocal neoplastic lesions were present, patients were qualified for mastectomy.
In accordance with your suggestions, data from tables 3 and 4 is now presented on charts.
Although we did not have a psychiatry specialist on our research team, patients from our healthcare facility were provided with psychological support by a psychological counselling team.
Thanks to the comments received, we were able to refine the publication in terms of the content as well as the language. All errors mentioned in the review were corrected in the final version of the publication.
Kind regards,
the Authors

Reviewer 2 Report
The paper is very interesting, particularly the use of the different QoL scales. Perhaps the point at which women who had MAS have the reconstructive surgery could be mentioned as this could affect the QoL results observed.
Lines 314-315 "The lowest results showing the best quality of life in symptomatic scales were 314 reported in the BCT+SLNB subgroup" are unclear.
More discussion and comparison of results to recent similar papers would be of added value, for example:
Psychiatr Pol. 2017 Oct 29;51(5):871-888. doi: 10.12740/PP/OnlineFirst/63787. Epub 2016 Oct 8.
Asian Pac J Cancer Prev 2014, 15(13):5377-81
Support Care Cancer 2020 Oct 24 doi:10.1007/s00520-020-05838-7.
Author Response
Dear Reviewer #2,
Thank you for reviewing our article titled ‘Quality of life in women subjected to surgical treatment of breast cancer depending on the procedure performed within the breast and axillary fossa — a single-center, one year prospective analysis’ We deeply appreciate your opinion as well as constructive comments that contributed to more profound consideration of issues addressed in our publication. The comments in your review will guide us in our future work.
In response to those commentaries we clarified the material and method section. In the course of our research, it was decided that patients who undergo breast reconstruction surgery after mastectomy should be excluded from this analysis. During this study, 33 patients underwent breast reconstruction surgery. Only 3 of those patients had simultaneous breast reconstruction, while the other 30 underwent this procedure at various times after their initial tumor removal surgery.
We improved lines 314-315 and clarifiedthem.
In accordance with your suggestions, we have further developed the discussion section by including some new research papers.
Thanks to the comments received, we were able to refine the publication in terms of the content as well as the language. All errors mentioned in the review were corrected in the final version of the publication.
Kind regards,
the Authors

Round 2
Reviewer 1 Report
Authors responded to previous issues. This reviewer finds while the authors do not present any practice changing or new data, the responses have been addressed. ok for publication
Author Response
Dear Reviewer,
Thank you for reviewing our article titled ‘Quality of life in women subjected to surgical treatment of breast cancer depending on the procedure performed within the breast and axillary fossa — a single-center, one year prospective analysis’ We deeply appreciate your opinion. We agree with you that we didn’t present any new data in revised version of our manuscript. Our study although is a single-center study describes very important topic- quality of life women after surgical treatment of breast. There are numerous articles discussing the quality of life of women who undergo surgical treatment for breast cancer - it has been a “trending” topic in the field for the past several years. However, our work stands out among other studies due to the fact that we introduced a year-long prospective evaluation of our patients.
We hope that you are finally accept our manuscript for publication.
Kind regards,
the Authors